# Investigation of synovial fluid induced *Staphylococcus aureus* aggregate development and its impact on surface attachment and biofilm formation

**Matthew J. Pestrak**[1], **Tripti Thapa Gupta**[1], **Devendra H. Dusane**[1], **Doug V. Guzior**[1], **Amelia Staats**[1], **Jan Harro**[2], **Alexander R. Horswill**[3], **Paul Stoodley**[1,4,5]*

1 Department of Microbial Infection and Immunity, The Ohio State University, Columbus, Ohio, United States of America, 2 Department of Microbial Pathogenesis, School of Dentistry, University of Maryland, Baltimore, Maryland, United States of America, 3 Department of Immunology and Microbiology, University of Colorado School of Medicine, Aurora, Colorado, United States of America, 4 Department of Orthopedics, The Ohio State University, Columbus, Ohio, United States of America, 5 National Centre for Advanced Tribology at Southampton (nCATS) and National Biofilm Innovation Centre (NBIC), Mechanical Engineering, University of Southampton, Southampton, United Kingdom

* Paul.Stoodley@osumc.edu

**Data Availability Statement:** All relevant data are within the manuscript and its Supporting Information files.

## Abstract

Periprosthetic joint infections (PJIs) are a devastating complication that occurs in 2% of patients following joint replacement. These infections are costly and difficult to treat, often requiring multiple corrective surgeries and prolonged antimicrobial treatments. The Gram-positive bacterium *Staphylococcus aureus* is one of the most common causes of PJIs, and it is often resistant to a number of commonly used antimicrobials. This tolerance can be partially attributed to the ability of *S. aureus* to form biofilms. Biofilms associated with the surface of indwelling medical devices have been observed on components removed during chronic infection, however, the development and localization of biofilms during PJIs remains unclear. Prior studies have demonstrated that synovial fluid, in the joint cavity, promotes the development of bacterial aggregates with many biofilm-like properties, including antibiotic resistance. We anticipate these aggregates have an important role in biofilm formation and antibiotic tolerance during PJIs. Therefore, we sought to determine specifically how synovial fluid promotes aggregate formation and the impact of this process on surface attachment. Using flow cytometry and microscopy, we quantified the aggregation of various clinical *S. aureus* strains following exposure to purified synovial fluid components. We determined that fibrinogen and fibronectin promoted bacterial aggregation, while cell free DNA, serum albumin, and hyaluronic acid had minimal effect. To determine how synovial fluid mediated aggregation affects surface attachment, we utilized microscopy to measure bacterial attachment. Surprisingly, we found that synovial fluid significantly impeded bacterial surface attachment to a variety of materials. We conclude from this study that fibrinogen and fibronectin in synovial fluid have a crucial role in promoting bacterial aggregation and inhibiting surface adhesion during PJI. Collectively, we propose that synovial fluid may have conflicting protective roles for the host by preventing adhesion to surfaces, but by promoting bacterial aggregation is also contributing to the development of antibiotic tolerance.

**Funding:** This work was supported by the National Institutes of Health grant R01GM124436 (PS).

**Competing interests:** The authors have declared that no competing interests exist.

## Introduction

Periprosthetic joint infections (PJIs) are a devastating complication of joint replacement surgeries, occurring in approximately 2% of all cases [1,2]. These infections often require multiple surgeries and extensive antibiotic treatments resulting in longer hospitalization and higher costs for the patient [2]. In addition to the economic burden associated with PJIs, nearly 26% of patients with PJIs requiring additional interventions die within 5 years of the initial surgery [3]. The Gram-positive bacterial species *Staphylococcus* are the most common cause of infection in these patients, accounting for nearly 45% of all PJIs [3,4]. *S. aureus* in particular is frequently isolated from these patients, and is often incredibly difficult to treat due to the development of antimicrobial tolerance [5]. *S. aureus* utilizes a number of strategies to impede antimicrobial killing and subvert the host immune system including, secreted proteases, surface factors, and biofilm development [5,6].

Biofilms are aggregated protective communities of bacteria that are surrounded by an extracellular matrix. The biofilm matrix is a complex structure of bacterial and host components consisting of extracellular DNA, proteins, and polysaccharides [7]. Once encased in the biofilm matrix, the bacteria exhibit enhanced tolerance to antimicrobials and the host immune system [8–10]. During PJI, *S. aureus* biofilms and aggregates have been observed on the surface of implanted joint devices and the surrounding tissue [11]. Although biofilms previously were defined as surface-associated communities, recent studies have demonstrated that bacterial aggregates function similarly protecting bacterial cells from the same external stressors [12–16]. Furthermore, bacterial aggregates have been observed in both wound and lung infections, indicating they have an important role during infection [17–20]. In the context of PJIs, *S. aureus* and *S. epidermidis* form dense aggregate structures in the presence of synovial fluid, a viscous lubricant present in the joint space [21,22]. Similar to surface associated biofilms, aggregation in synovial fluid provides the bacteria with enhanced tolerance to antimicrobial treatment and phagocytosis [12,23,24]. Therefore, it is essential to understand how synovial fluid promotes aggregation and influences the establishment of chronic infections.

A role for a number of *S. aureus* factors in this process have been identified [23], but many aspects *S. aureus* aggregate formation in synovial fluid remain unclear. In this study, we examined the kinetics of aggregate formation in synovial fluid and identified which host factors are involved in this process. Utilizing flow cytometry to quantify aggregate formation, we determined that *S. aureus* aggregates within minutes of synovial fluid exposure. Furthermore, we determined this process is mediated predominately by *S. aureus* interaction with host fibrinogen and fibronectin. Finally, as surface-associated biofilms have been observed on implants during PJI, we investigated the effects of synovial fluid on *S. aureus* surface adhesion. Interestingly, synovial fluid drastically inhibited surface attachment to plastic, glass, titanium, steel, and hydroxyapatite. Collectively, we propose that synovial fluid may have conflicting protective roles for the host by preventing adhesion to surfaces, but by promoting bacterial aggregation is also contributing to the development of antibiotic tolerance.

## Methods

### Bacterial growth conditions and bacterial strains

In all experiments, *S. aureus* was grown in tryptic soy broth at 37˚C in a shaking incubator operating at 200 RPM for 17–18 hours. The coupon adherence assays were completed with the GFP tagged *S. aureus* strain AH1726 [25]. All other experiments were completed using the clinical *S. aureus* isolate CGS.Sa03 [26].

## Bacterial aggregate quantification

The optical density at 600 nm was measured for each culture and 0.75 $OD_{600}$ of cells from stationary phase cultures was pelleted at 21,000 xg and suspended in 1 mL of Ringer's solution buffer (BR0052G, Fisher Scientific). Cells were stained with SYTO9 (Invitrogen, Thermo Fisher Scientific, Waltham MA, USA) for 10 min at room temperature and washed three times in Ringer's solution. Next, the cells were suspended in 500 μl of Ringer's solution or 10% bovine synovial fluid (Lampire Biological Laboratories, Pipersville,PA, USA) in Ringer's solution. For treatments with purified synovial fluid components, 19 mg/ml of BSA (Fisher, BP1600), 1 mg/ml of DNA (Ambion Salmon Sperm DNA, AM9680) (to mimic native circulating cell free DNA (ccfDNA)), 3 mg/ml of hyaluronic acid (Fisher, AAJ60566MA), 0.172 mg/ml of human fibrinogen (Invitrogen, PIRP43142), or 450 ug/ml of fibronectin (Alfa Aesar, BT 226) was added to bacterial suspensions which are in the range of these components reported in patient arthritic knee synovial fluid [27–31]. Cells were then incubated at room temperature for 5 to 120 minutes as indicated. Following incubation, 100 μl of the cells was collected from the bottom of the microcentrifuge tube and transferred slowly to a 5 ml round bottom polystyrene tube. Single bacterial cells can be differentiated from cell aggregates using flow cytometry [32], so we quantified bacterial aggregation using a BD FACsCanto II flow cytometer (BD sciences), as previously described [14]. The forward and side scatter of the SYTO9+ population was quantified to exclude unstained synovial fluid debris and quantify only the bacterial population. All flow cytometry data was quantified using FlowJo 9.0. The population of single cells was determined by gating a population single bacterial cells in the negative control confirmed by light microscopy. The percentage of the population existing as aggregates was calculated by subtracting the single celled population from the total population. To determine the average size of aggregates within a population, the median fluorescence intensity (MFI) of the forward scatter was calculated. At least 10,000 events were measured for each sample in triplicate in at least two independent experiments. Statistical significance was determined by Student's T-test or one-way ANOVA followed by Dunnett's multiple comparison test to compare means against the untreated control when applicable.

## Confocal microscopy

250μl of stationary phase cultures were pelleted, washed, and suspended in 250 μl of Ringer's solution. Cells were stained with SYTO9 for 10 minutes at room temperature. Stained cells were washed three times and suspended in 250 μl of Ringer's solution. 50 μl of cells were suspended in 600 μl of Ringer's solution or 10% bovine synovial fluid in Ringer's solution. Cells were incubated at room temperature in confocal cover-glass bottom petri dishes for 5–60 minutes allowing for aggregation to occur. Imaging of cells and aggregates were taken at multiple time points under 60x magnification using a Olympus FluoView FV10i Confocal Laser Scanning Microscope.

## Quantification of bacterial surface attachment

For both *S. aureus* strains, 0.75 $OD_{600}$ of overnight stationary phase cultures were pelleted by centrifugation at 21,000 xg for 1 minute and suspended in 500 μl of Ringer's solution or 10% synovial fluid in Ringer's solution. The cells were incubated for 30 minutes at room temperature and then diluted in 50 ml of Ringer's solution. A peristaltic pump was used to flow the cultures through a 6-well IBIDI flow cell with a constant shear stress of 8.4 mPa unless otherwise specified. Using an inverted epifluorescence microscope (EVOS, Thermo Fisher Scientific Inc., Waltham MA USA), three images were taken per channel and averaged together for each condition after 5 minutes. All experiments were completed in triplicate in at least two

independent experiments, and statistical significance was determined by two-way ANOVA followed by Tukey's multiple comparisons test.

To measure bacterial adhesion to different surface types, overnight cultures of GFP producing *S. aureus* cells were centrifuged at 21,000 xg for 1 minute. The supernatant was removed, and the pellet was washed in PBS and suspended in 10% synovial fluid in ringer's solution. The suspension was then incubated for 1 hour to form aggregates. The attachment of single cells and aggregates under flow was observed using a Leica DM2700 M upright microscope on different coupon materials: Titanium (Ti), stainless steel (316L), and hydroxyapatite (HA) (10 mm diameter and 2mm thickness; BioSurface Technologies) using a 20X objective. To ensure similar roughness, all coupons were sanded using Grainger P600 aluminum oxide sandpaper for 4–5 minutes. Using a peristaltic pump, the bacteria were pumped through the flow cell at a constant shear stress of 15 mPa for 5 minutes. Time lapse videos were recorded at 30 fps using Micromanager software and QIClick CCD digital camera. Using ImageJ, ten frames were analyzed and the number of attached bacterial particles was quantified after 5 minutes. A threshold was applied to each video and the average intensity of GFP signal was quantified across all frames. Particles with a low average intensity indicated the bacteria did not adhere and was present the liquid phase. Therefore, low intensity particles were excluded, and high intensity attached particles were quantified to determine how much bacteria was present on the coupon surface. All experiments were done in triplicate and statistical significance was determined by Student's T-test.

## Proteinase K and heat treatment of synovial fluid

To disrupt proteins in the synovial fluid 250 µg/ml of proteinase K was added to 1 ml of synovial fluid and incubated at 37˚C for one hour. For heat treatment, 1 ml of synovial fluid was boiled at 100˚C for 30 minutes. After protein disruption cells were treated with synovial fluid for 30 minutes as described previously.

## Statistical analysis

Prism (Graphpad v7.04 software) was used for all statistical analysis. The threshold for significance was set at P value < 0.05. Statistically significant differences were determined using the test specified in the corresponding methods sections. All error bars indicate standard error of the mean.

## Results

### Synovial fluid induces *S. aureus* aggregation

Flow cytometry was utilized to assess *S. aureus* aggregate formation of the clinical isolate CGS. Sa03 following exposure to synovial fluid. Compared to the untreated, single cell control culture, synovial fluid induced the formation of large bacterial aggregates as indicated by increased average forward and side scatter (Fig 1A–1D). While the percent population indicates the relative number of aggregates compared to single cells, it does not provide an indication of aggregate size. In order to better assess particle sizes, the median fluorescence intensity (MFI) was calculated for each population (Fig 1C and 1D). We observe increased MFI following synovial fluid exposure indicating particle size was increased. Finally, these observations were confirmed using light microscopy. As expected, cells in the untreated controls existed predominately as single cells, while synovial fluid treated cultures contained many large aggregated bacterial clusters (Fig 1A).

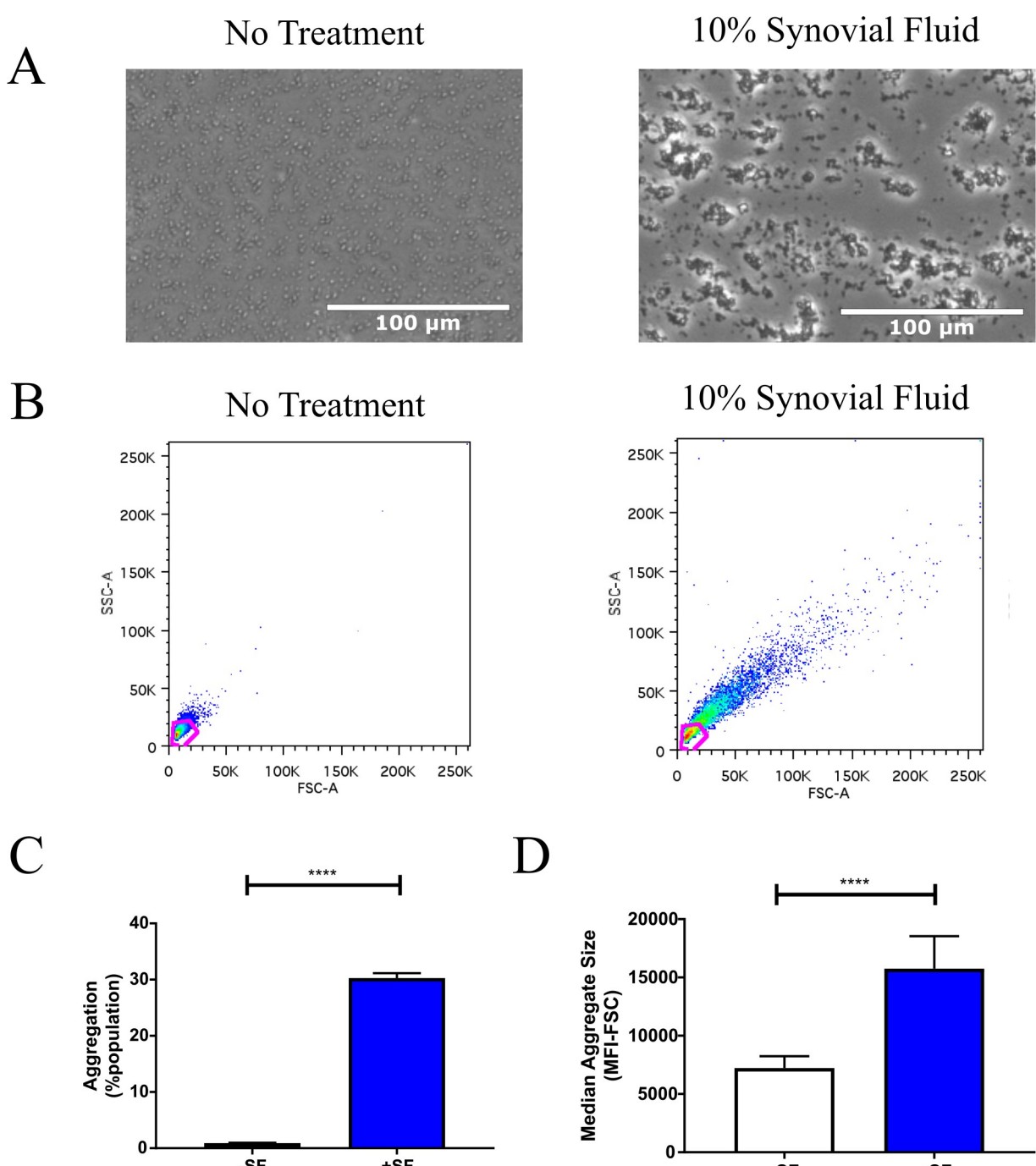

**Fig 1. Synovial fluid promotes *S. aureus* aggregation. A**) Image of CGS.Sa03 in ringer's solution +/- 10% synovial fluid. **B**) Flow cytometry was used to determine the aggregation index of CGS.Sa03 in 10% synovial fluid after 1 hour of incubation. **CD**) The median forward scatter signal intensity was quantified as an indicator of the relative size of aggregates in a given population. Error bars indicate mean ± SEM. Statistical significance was determined by Student's T-test. ****p<0.0001.

### Aggregates form rapidly in synovial fluid

Considering aggregation promotes antimicrobial tolerance, it is important to understand how quickly these aggregates form during an infection. Therefore, we sought to determine the rate

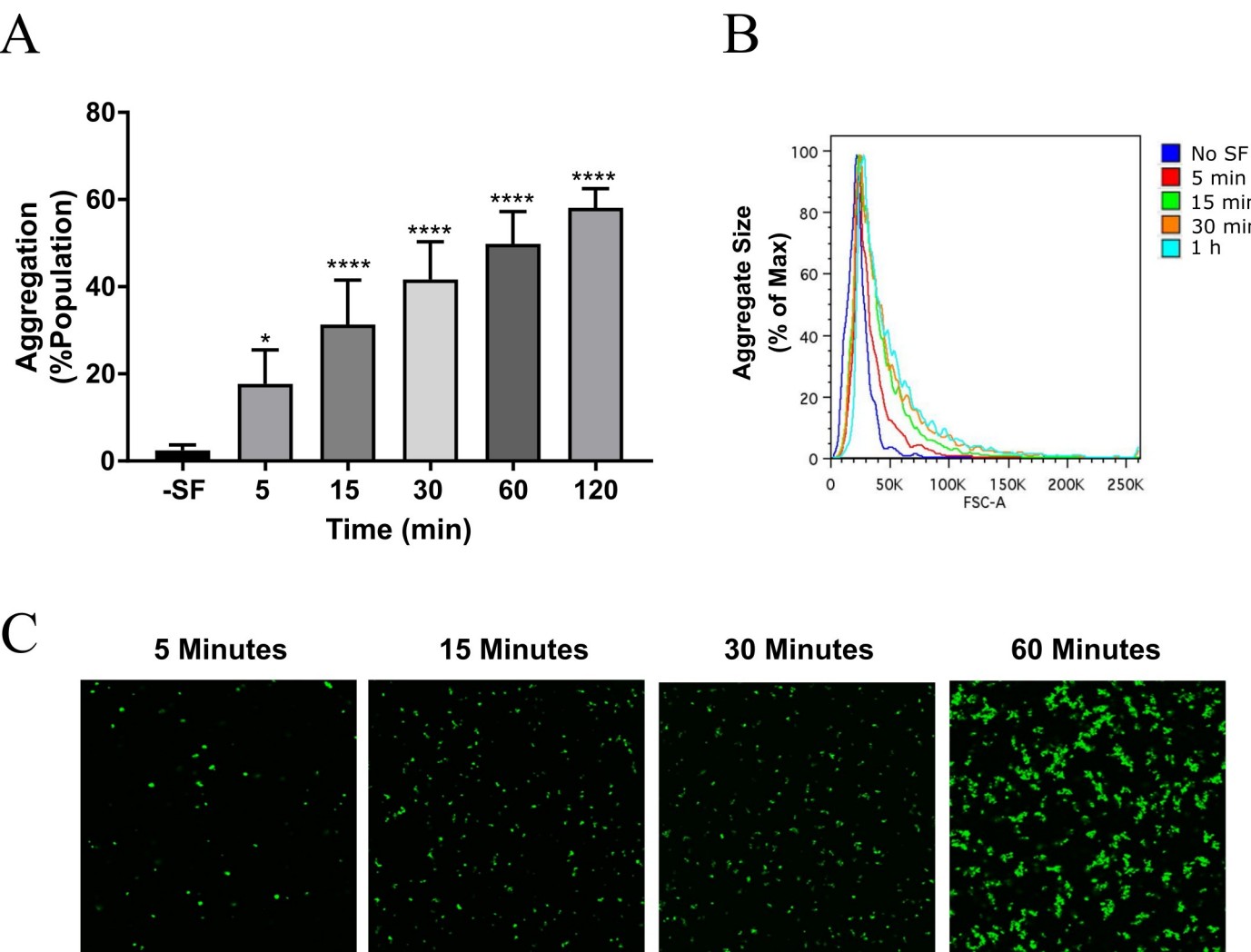

**Fig 2. Synovial fluid promotes aggregation in a cell concentration and time-dependent manner. AB)** *S. aureus* cells were treated with 10% synovial fluid in Ringer's solution and flow cytometry was used to quantify the aggregation index at the indicated times over a two hour period. **C)** Aggregate formation was observed using confocal microscopy. Error bars indicate mean ± SEM. Statistical significance was determined by one-way ANOVA followed by Dunnett's multiple comparison test to compare means against the untreated control. *p<0.05, ****p<0.0001.

of aggregation formation in synovial fluid using flow cytometry and confocal microscopy. After 15 minutes, nearly 40% of the bacterial population was incorporated into an aggregate structure (Fig 2A). The percentage of aggregates in the population appeared to continue increasing over 60 minutes (Fig 2B), but this rate appeared to level off after a certain amount of time. This could indicate that *S. aureus* aggregates reached a maximum size and could no longer incorporate additional cells, due to either physical limitations or quorum-sensing mediated biofilm dispersal mechanisms [33]. Alternatively, we did not observe any aggregation above 60% in our studies, which may indicate an upper limit of detection with this method.

These results were further confirmed with confocal microscopy by imaging aggregate development of SYTO9 stained *S. aureus* cells in the presence of 10% synovial fluid (Fig 2C). Aggregate formed rapidly and were visible after 15 minutes. Therefore, we conclude that *S. aureus* will form aggregates within minutes of contact with synovial fluid exposure. Considering aggregates provides the bacteria with protection for immune clearance and drug treatment, we

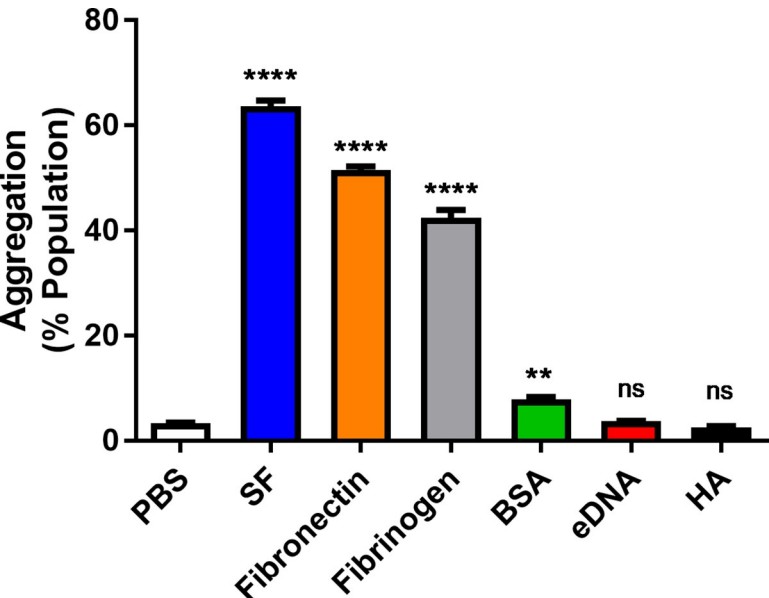

**Fig 3. Fibronectin contributes to *S. aureus* aggregation.** Aggregation was quantified using flow cytometry following 30 minute treatment with physiologically relevant concentrations of the indicated synovial fluid components. Error bars indicate mean ± SEM. Statistical significance was determined by one-way ANOVA followed by Dunnett's multiple comparison test to compare means against the untreated control. **p<0.01, ****p<0.0001.

anticipate the rapid nature this process may have a crucial role in the establishment of PJI infections.

## Fibronectin and fibrinogen are sufficient to induce *S. aureus* aggregation

While it is now established that synovial fluid promotes aggregate formation [21], it remains unclear which components in synovial fluid promote aggregation. A study by Dastgheyb *et al.* determined that *S. aureus* transposon mutants deficient in binding fibrinogen and fibronectin aggregated poorly in synovial fluid [12]. Similarly, *S. aureus* aggregation has been observed in serum due to the interaction between surface receptors and fibrinogen [20,34]. Another study determined that hyaluronic acid promoted aggregation in strains lacking hyaluronidase production [22]. Taken together, these studies indicate *S. aureus* interaction with host factors may be important for aggregation in synovial fluid, but it remains unclear specifically which synovial fluid factors promote *S. aureus* aggregation. To better elucidate which factors are sufficient to cause aggregation, we treated *S. aureus* with purified synovial components at concentrations similar to the observed level in the human joint space [27,28,30,35]. In agreement with previous reports, we observe that both fibrinogen and fibronectin are sufficient to promote *S. aureus* aggregation (Fig 3). Additionally, we observe slightly increased aggregation in the presence of serum albumin. While high concentrations of hyaluronic acid (3 mg/ml) promotes *S. aureus* aggregation in strains lacking hyaluronidase [22], we did not observe significant aggregation of CGS.Sa03 following treatment with hyaluronic acid. Similarly, cell free DNA did not appear to stimulate aggregation. These data indicate that hyaluronic acid and cell free host DNA may not have a crucial role in synovial fluid mediated aggregate development in some clinical strains of *S. aureus*. One limitation of this experiment is that we used DNA purified from Salmon sperm, which is likely different than the cell free DNA found in synovial fluid. Furthermore, *S. aureus* produces a number of nucleases that degrade extracellular DNA [36,37]. While cell free DNA and hyaluronic acid did not cause CGS.Sa03 to aggregate, these

factors may have an important role in aggregation for strains or growth conditions leading to low nuclease or hyaluronidase production. That being said, fibronectin and fibrinogen were sufficient to induce aggregation to levels similar to 10% synovial fluid for the clinical strain CGS. Sa03. This likely indicates these factors have a major role in aggregate formation during PJI.

## Synovial fluid aggregation inhibits bacterial surface attachment

Biofilms form on the surface of implanted medical devices and prosthetic joint components [10,11], but how this process occurs during PJI remains unclear. The initiation of biofilm formation first requires bacteria to adhere to the surface. Considering free-floating aggregates rapidly form after contact with synovial fluid (Fig 2), we hypothesized these aggregates could function as proto-biofilms that adhere to the implant and transition to a surface-associated biofilm. Therefore, to determine how synovial fluid affects surface attachment, we measured CGS.Sa03 cell surface attachment to plastic under various shear stresses. While the exact shear stresses and fluid movement in the joint space has not been reported, we expect a range of stresses would be present depending on joint activity and the relative location within the joint. To replicate the conditions within the joint, we examined attachment under various flow conditions. Using shear stresses between 0.77–816 mPa, attachment was assessed ranging from nearly static conditions to stresses similar to the human artery [38]. Unexpectedly, we observe a significant decrease in *S. aureus* attachment following synovial fluid exposure regardless of the shear stress (Fig 4).

## Synovial fluid inhibits attachment to multiple surface types

Joint implants typically consist of multiple components and are often made out of polyethylene, titanium, steel, and cobalt-chromium, which are then cemented into place with hydroxyapatite. To determine if synovial fluid inhibits attachment to surfaces other than plastic, we utilized a BioSurface flow cell system with coupon inserts of titanium, stainless steel (alloy 316L), and hydroxyapatite. Regardless of the material, synovial fluid significantly reduces the ability of *S. aureus* AH1726 to adhere to a surface (Fig 5). These data suggest this phenotype is not specific to just one surface type and synovial fluid likely inhibits attachment to the implant during PJI.

## Synovial fluid components inhibit bacterial surface attachment

To better understand how synovial fluid inhibits *S. aureus* surface attachment, CGS.Sa03 cells were treated with purified components of synovial fluid at physiologically relevant concentrations [27,28,30,35]. We observe reduced *S. aureus* attachment following treatment with fibronectin, fibrinogen, and serum albumin, but not after treatment with cell free DNA and hyaluronic acid (Fig 6A). To confirm that synovial fluid proteins are responsible for decreased surface attachment, synovial fluid proteins were degraded prior to bacterial treatment with heat or proteinase K treatment. In both cases, we see partially restored bacterial surface adhesion, indicating that these protein factors are inhibiting surface attachment (Fig 6B). Based on our flow cytometry analysis, fibronectin, fibrinogen, and to a lesser extent BSA all promoted aggregation (Fig 3). This could suggest that bacterial aggregation limits surface attachment. One explanation for this observation could be that increased particle size leads to higher drag forces reducing attachment.

## Discussion

We have previously reported that bacterial aggregates are present during periprosthetic joint infection [11]. Nearly 50% of *S. aureus* PJI infections exhibit antibiotic tolerance, and the

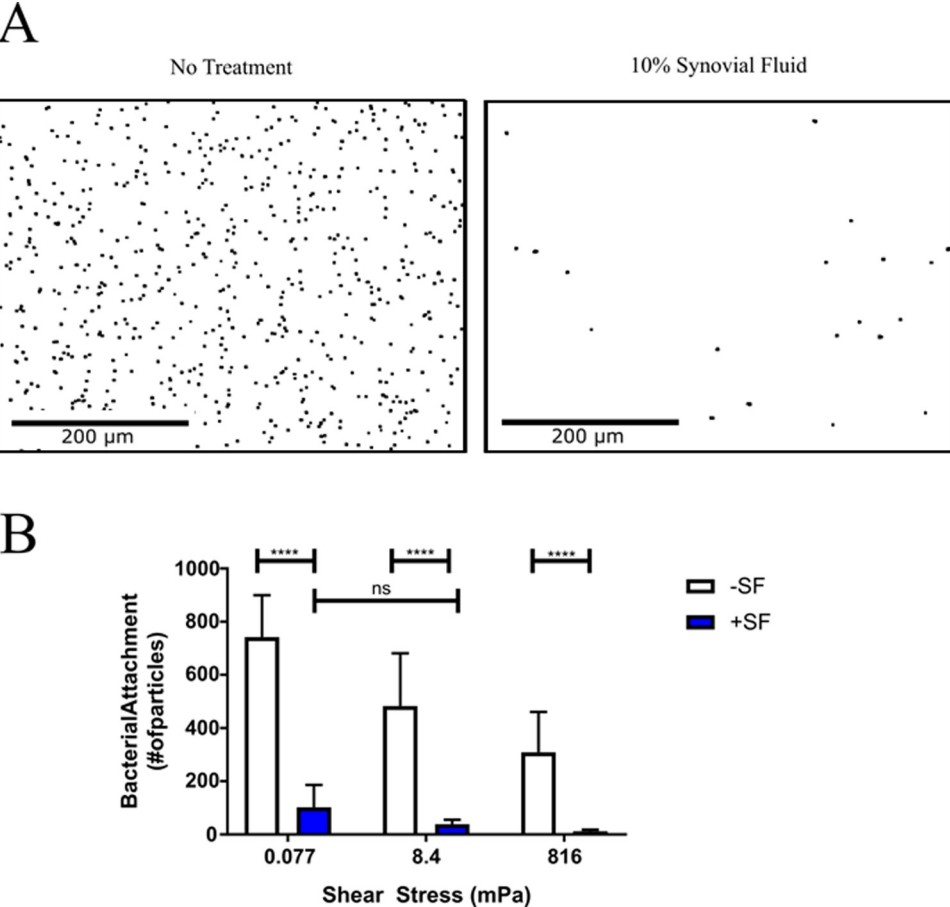

**Fig 4. Synovial fluid aggregation inhibits bacterial surface attachment. A**) Representative images of CGS.Sa03 surface attachment and quantification (**B**) after 5 minutes in flow conditions (mPa 8.4). Error bars indicate mean ± SEM. Statistical significance was determined by two-way ANOVA followed by Tukey's multiple comparisons test. ****p<0.0001.

development of *S. aureus* aggregates enhances tolerance towards antibiotics and the host immune system [6,12,23,24,39–41]. Herein, we utilized a novel method for precise quantification of synovial fluid induced aggregation using flow cytometry (Figs 1–3). In agreement with previous studies [13,21,22], we observe that *S. aureus* aggregates readily in synovial fluid (Fig 1). Furthermore, we demonstrate this process occurs within minutes (Fig 2), which could have important implications for single cells entering the joint space following surgery. Based on these findings, we anticipate synovial fluid has a crucial role in the establishment of PJI infections, and that preventing aggregate formation may be an effective strategy for preventing *S. aureus* colonization and improving antimicrobial efficacy.

Currently, chemical and enzymatic therapeutics for biofilm dispersion is a major focus of drug development, which includes proteases and DNases [5,42–44]. These dispersal agents have shown promise in treating infections for a variety of bacterial species including *S. aureus* [44–47]. Herein, we report that fibrinogen or fibronectin was sufficient to generate large *S. aureus* aggregates. Since the aggregate matrix appears to be predominately composed of host factors, it may be difficult to develop therapeutics directly targeting these structures. However, disrupting *S. aureus* surface factors that bind to fibrinogen and fibronectin may be an effective alternative for preventing aggregate formation. *S. aureus* produces a number of structurally

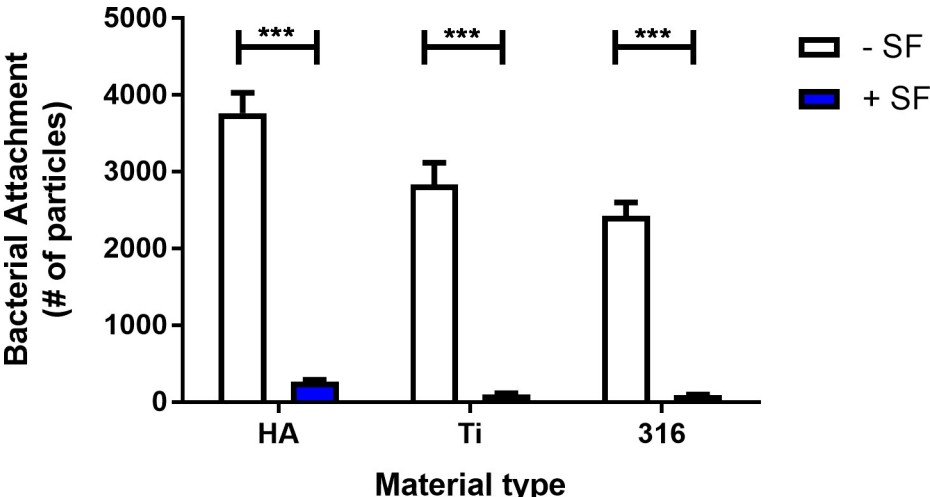

**Fig 5. Synovial fluid inhibits attachment to various orthopedic material.** *S. aureus* surface attachment to the titanium (Ti), hydroxyapatite (HA), and stainless steel (316L) was quantified after 5 minutes under constant shear tress of 15 mPa. Error bars indicate mean ± SEM. Statistical significance was determined by Student's T-test. ***p<0.001.

similar surface factors known as microbial surface components recognizing adhesive matrix molecules (MSCRAMMs). As their name suggests, these factors have an important role in surface adhesion, but they are also important for host immune evasion, host cell invasion, and biofilm formation [48,49]. In the context of synovial fluid aggregation, loss of functional FnbA, FnbB, ClfA, and ClfB resulted in decreased aggregate size [23,50]. In addition to these

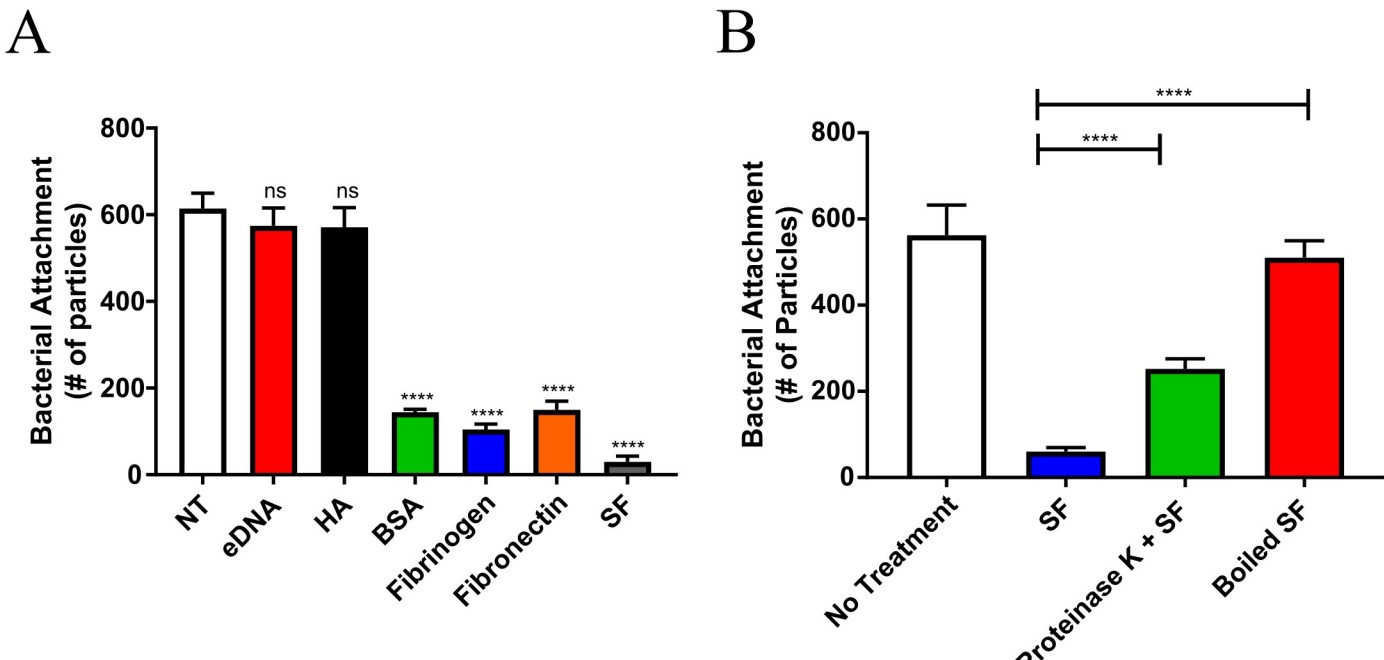

**Fig 6. The protein components in synovial fluid impede bacterial surface attachment. A)** *S. aureus* was treated with the indicated synovial fluid component and bacterial attachment was quantified after 5 minutes of flow (8.4mPa). **B)** Bacteria was treated with boiled synovial fluid and surface attachment was quantified. Error bars indicate mean ± SEM. Statistical significance was determined by one-way ANOVA followed by Dunnett's multiple comparison test to compare means against the untreated control (A) or Tukey's multiple comparisons test (B). ****p<0.0001.

surface factors role during infection, they all directly bind to fibrinogen and fibronectin [20]. Considering we observe substantial aggregation in purified fibrinogen and fibronectin (Fig 3), these data provide further evidence that *S. aureus* interaction with host factors has an important role in aggregate and biofilm formation during infection.

Abiotic implant surfaces in the host environment are rapidly coated in a "conditioning film" of host proteins, which includes fibronectin and fibrinogen [51,52]. During infection, it is thought that bacterial cells directly interact with the conditioning film rather than the implant's surface [1,51]. Thus, MSCRAMMs likely have an important role in surface adhesion by binding to host factors in the conditioning film. Interestingly, we observed a significant decrease in *S. aureus* surface adhesion following exposure to synovial fluid (Figs 4 & 5), which could be partially restored by degrading synovial fluid proteins (Fig 6). Considering *S. aureus* binds to fibrinogen and fibronectin, reduced attachment may be due to synovial fluid sequestering surface receptors or steric hindrance that are required for surface adhesion. A previous study demonstrated the risk of infection is reduced if bacterial adhesion to an implant is delayed [53]. Therefore, synovial fluid may have a protective role for the host by preventing *S. aureus* from binding to the implant surface and subsequently inhibiting biofilm formation.

Considerable work has been done studying bacterial infections in the context of planktonic single cells and surface associated biofilms. In the conventional biofilm model, a single bacterial cell adheres to an inert surface and develop into a surface associated biofilm [7,9]. While this may often occur during infection, recent studies have identified the presence of large bacterial aggregates during infection [19,54]. In addition to synovial fluid mediated aggregation, there is evidence suggesting aggregated bacterial clusters disseminate from mature biofilms to seed new areas during growth [55,56]. Collectively, these studies suggest aggregates have a major role during infection and in biofilm development. Our findings here further demonstrate that studying aggregates in the context of infection will be necessary to fully understand how chronic infections develop. Based on the paradigm that single cells initiate biofilm formation, a multitude of studies have addressed how single cells attach to surfaces. As we continue to develop our understanding of the role aggregates play during infection, it may be necessary to reevaluate which factors are important to consider when predicting if an implant will become infected. Considering we observe minimal attachment to surfaces in the presence of synovial fluid (Figs 4 & 5), it could indicate larger scale features, such as crevices or grooves, in the implant or tissue are more relevant than individual cell interactions for trapping aggregates in place. While we predict synovial fluid mediated aggregation has a key role in the development of biofilms during PJI, we anticipate this role is limited to infections at sites where synovial fluid is present. Thus, *S. aureus* likely utilizes different mechanisms for biofilm formation depending on the body site, as emphasized by the various biofilm structures observed during different types of infection, such as *Staphylococcus* abscess communities [33,57].

In conclusion, given the difficult nature of treating biofilm infections, methods for preventing and treating these infections has become increasingly important as multi-drug resistance becomes more common. Although still largely in the preclinical stages of development, efficacy has been demonstrated for bandage and implant coatings that prevent bacterial colonization and promote host cell attachment on implant materials [5,6,58–60]. While we acknowledge that future *in vivo* studies will be necessary to fully understand the extent of synovial fluid's role during PJI, our study provides evidence that *S. aureus* interaction with synovial fluid has an important role during PJI. We report that synovial fluid promotes bacterial aggregation, while simultaneously impeding surface attachment. While preventing adhesion to the implant surface is beneficial to the host, the formation of aggregates and subsequent antibiotic tolerance can be detrimental. Therapeutics that merely prevent aggregation in synovial fluid may result in increased adhesion and biofilm formation. While we anticipate disrupting *S. aureus*

aggregates will be crucial improving antimicrobial efficacy, it may be necessary to additionally focus on strategies that inhibit adhesion to surfaces in the host environment.

## Supporting information

**S1 Data.**
(XLSX)

## Author Contributions

**Conceptualization:** Matthew J. Pestrak, Tripti Thapa Gupta, Devendra H. Dusane, Doug V. Guzior, Amelia Staats, Jan Harro, Alexander R. Horswill, Paul Stoodley.

**Data curation:** Matthew J. Pestrak, Tripti Thapa Gupta.

**Formal analysis:** Matthew J. Pestrak, Tripti Thapa Gupta, Devendra H. Dusane, Doug V. Guzior, Amelia Staats, Jan Harro, Alexander R. Horswill, Paul Stoodley.

**Funding acquisition:** Paul Stoodley.

**Investigation:** Matthew J. Pestrak, Tripti Thapa Gupta, Devendra H. Dusane, Doug V. Guzior, Amelia Staats, Jan Harro, Alexander R. Horswill, Paul Stoodley.

**Methodology:** Matthew J. Pestrak, Devendra H. Dusane, Amelia Staats, Jan Harro, Alexander R. Horswill, Paul Stoodley.

**Project administration:** Paul Stoodley.

**Supervision:** Paul Stoodley.

**Writing – original draft:** Matthew J. Pestrak, Tripti Thapa Gupta, Devendra H. Dusane.

**Writing – review & editing:** Doug V. Guzior, Amelia Staats, Jan Harro, Alexander R. Horswill, Paul Stoodley.

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
