## [Decision Letter · Decision Letter 0]

3 Mar 2020

PONE-D-20-01923

Investigation of synovial fluid induced Staphylococcus aureus aggregate development and its impact on surface attachment and biofilm formation

PLOS ONE

Dear Dr. Paul Stoodley,

Thank you for submitting your manuscript to PLOS ONE. After careful consideration, we feel that it has merit but does not fully meet PLOS ONE’s publication criteria as it currently stands. Therefore, we invite you to submit a revised version of the manuscript that addresses the points raised during the review process.

We would appreciate receiving your revised manuscript by Apr 17 2020 11:59PM. To enhance the reproducibility of your results, we recommend that if applicable you deposit your laboratory protocols in protocols.io, where a protocol can be assigned its own identifier (DOI) such that it can be cited independently in the future. For instructions see: http://journals.plos.org/plosone/s/submission-guidelines#loc-laboratory-protocols

We look forward to receiving your revised manuscript.

Kind regards,

Abdelwahab Omri, Pharm B, Ph.D

Academic Editor

PLOS ONE

Journal Requirements:

Reviewers' comments:

Reviewer's Responses to Questions

**Comments to the Author**

1. Is the manuscript technically sound, and do the data support the conclusions?

Reviewer #1: Yes

Reviewer #2: Yes

2. Has the statistical analysis been performed appropriately and rigorously? 

Reviewer #1: Yes

Reviewer #2: Yes

3. Have the authors made all data underlying the findings in their manuscript fully available?

Reviewer #1: Yes

Reviewer #2: Yes

4. Is the manuscript presented in an intelligible fashion and written in standard English?

Reviewer #1: Yes

Reviewer #2: Yes

5. Review Comments to the Author

Reviewer #1: This manuscript reports a study that sheds new light on prosthetic joint infections with S. aureus biofilms. The role of synovial fluid in reducing biofilm formation through the formation of cell aggregates has not been reported previously and is somewhat unexpected. Aggregation of bacteria is often associated with biofilm formation but that relates to interactions between the cells without the involvement of extrinsic factors such as synovial fluid. The observation in this manuscript is interesting in that it suggests a natural defence mechanism to reduce biofilm formation on prosthetic joint materials. As a reviewer I am not familiar with the amount of synovial fluid present during prosthetic joint surgery but this manuscript suggests that the presence of synovial fluid should be encouraged. The effect of cell aggregation in preventing biofilm formation may be due to a number of factors including synovial fluid sequestering surface receptors or the size of the aggregates, initiating forces that prevent the binding of these aggregates to surface. The authors have discussed these possibilities. Some study of the aggregates - there structure, chemistry and physical nature will be important. This may lead to novel biofilm preventative measures in other environments.

The issue of antibiotic resistance of the aggregates is something that needs to be investigated.

The manuscript is well written. Only a few minor - recommendations and these all relate to the need for bacterial names to be italicized in the reference list. References 14, 20 and 40 all have bacterial names that have not been italicized.

Reviewer #2: The manuscript by Pestrak et al entitled “Investigation of synovial fluid induced Staphylococcus aureus aggregate development and its impact on surface attachment and biofilm formation” is a well-focused in vitro study by experts in the field. As it is current unknown how PJI biofilm formation commences, which is a significant question from both a scientific and clinical standpoint, the authors addressed this in vitro by demonstrating that fibrinogen and fibronectin promoted bacterial aggregation, while cell free DNA, serum albumin, and hyaluronic acid had minimal effect. Additionally, they used microscopy to measure bacterial attachment to implants and found that synovial fluid significantly impeded bacterial surface attachment to a variety of materials. Based on this, the authors conclude that: aggregation in synovial fluid is very rapid, has a threshold of ~60%, fibrinogen and fibronectin in synovial fluid have a crucial role in promoting bacterial aggregation and inhibiting surface adhesion during PJI, and that synovial fluid may have conflicting protective roles for the host by preventing adhesion and promoting bacterial aggregation. As their results support these novel conclusion, the manuscript is considered to be an important advance in this field. However, there are a few points that the authors should address.

1) The authors cite dated literature in the Introduction, which could be markedly improved.

Current data on the incidence and costs of PJI have recently been reported [1], and there is new consensus information of biofilm formation in PJI [2].

2) While the authors’ focused study on synovial fluid aggregates and implant biofilm formation is justified, it is now know that S. aureus utilizes four distinct mechanisms of biofilm formation during PJI [3]. Thus, the lack of studies on Staphylococcus abscess communities (SACs) and colonization of osteocytic canalicular networks in cortical bone need to be acknowledged as limitations.

3) The absence of in vivo studies should also be acknowledge as a limitation.

4) The authors conclude that the aggregation threshold of 60% in their studies is due to a size limit or an upper limit of detection with their method. However, this could also be due to Agr mediated emigration.

1. Schwarz EM, Parvizi J, Gehrke T, Aiyer A, Battenberg A, Brown SA, et al. 2018 International Consensus Meeting on Musculoskeletal Infection: Research Priorities from the General Assembly Questions. J Orthop Res. 2019;37(5):997-1006. Epub 2019/04/13. doi: 10.1002/jor.24293. PubMed PMID: 30977537.

2. Saeed K, McLaren AC, Schwarz EM, Antoci V, Arnold WV, Chen AF, et al. 2018 international consensus meeting on musculoskeletal infection: Summary from the biofilm workgroup and consensus on biofilm related musculoskeletal infections. J Orthop Res. 2019;37(5):1007-17. Epub 2019/01/23. doi: 10.1002/jor.24229. PubMed PMID: 30667567.

3. Schwarz EM, McLaren AC, Sculco TP, Brause B, Bostrom M, Kates SL, et al. Adjuvant Antibiotic-Loaded Bone Cement: Concerns with Current Use and Research to Make it Work. J Orthop Res. 2020. Epub 2020/01/31. doi: 10.1002/jor.24616. PubMed PMID: 31997412.

6. PLOS authors have the option to publish the peer review history of their article (what does this mean?). If published, this will include your full peer review and any attached files.

Reviewer #1: Yes: Steve Flint

Reviewer #2: No

---

## [Author Response · Author response to Decision Letter 0]

26 Mar 2020

Dear Dr. Abdelwahab Omri, 

Please find uploaded our revised manuscript, PONE-D-20-01923 “Investigation of synovial fluid induced Staphylococcus aureus aggregate development and its impact on surface attachment and biofilm formation.”

We have responded to each of the critiques outlined by the reviewers. We found the reviews to

be accurate and helpful in the development of this manuscript and sincerely appreciate the

constructive comments, which have helped to clarify several important points in the manuscript

and present the data more clearly and concisely to the readers. We trust that the corrections

made to the manuscript make it acceptable for publication.

Below is an item-by-item response to the critique:

Reviewer #1

Comment: Only a few minor - recommendations and these all relate to the need for bacterial names to be italicized in the reference list. References 14, 20 and 40 all have bacterial names that have not been italicized.

Response: The bacterial names in the references have been italicized. 

Reviewer #2: 

Comment: The authors cite dated literature in the Introduction, which could be markedly improved. Current data on the incidence and costs of PJI have recently been reported [1], and there is new consensus information of biofilm formation in PJI [2].

Response: We thank the reviewer for this correction, and we have included these references and updated the text accordingly (see Lines 59, 61, & 76)

Comment: While the authors’ focused study on synovial fluid aggregates and implant biofilm formation is justified, it is now known that S. aureus utilizes four distinct mechanisms of biofilm formation during PJI [3]. Thus, the lack of studies on Staphylococcus abscess communities (SACs) and colonization of osteocytic canalicular networks in cortical bone need to be acknowledged as limitations.

Response: We agree with the reviewer that this is a limitation of our study, and discussion of this limitation has been included in the discussion (see Lines 335-339). 

Comment: The absence of in vivo studies should also be acknowledged as a limitation.

Response: We also agree that this is another limitation of the study, and discussion of this has been included in the discussion (see Lines 344-346). 

Comment: The authors conclude that the aggregation threshold of 60% in their studies is due to a size limit or an upper limit of detection with their method. However, this could also be due to Agr mediated emigration.

Response: We agree that this is another possible explanation for our observations, and we have included this interpretation in the results section (see Lines 199-200).

---

## [Editor Report · Decision Letter 1]

1 Apr 2020

Investigation of synovial fluid induced Staphylococcus aureus aggregate development and its impact on surface attachment and biofilm formation

PONE-D-20-01923R1

Dear Dr. Paul Stoodley,

We are pleased to inform you that your manuscript has been judged scientifically suitable for publication and will be formally accepted for publication once it complies with all outstanding technical requirements.

With kind regards,

Abdelwahab Omri, Pharm B, Ph.D

Academic Editor

PLOS ONE

---

## [Editor Report · Acceptance letter]

6 Apr 2020

PONE-D-20-01923R1 

Investigation of synovial fluid induced Staphylococcus aureus aggregate development and its impact on surface attachment and biofilm formation 

Dear Dr. Stoodley:

I am pleased to inform you that your manuscript has been deemed suitable for publication in PLOS ONE. Congratulations! Your manuscript is now with our production department. 

With kind regards,

on behalf of

Dr. Abdelwahab Omri 

Academic Editor

PLOS ONE